# Cancer in children born after frozen-thawed embryo transfer: A cohort study

Nona Sargisian[1], Birgitta Lannering[2], Max Petzold[3], Signe Opdahl[4], Mika Gissler[5,6], Anja Pinborg[7], Anna-Karina Aaris Henningsen[7], Aila Tiitinen[8], Liv Bente Romundstad[9,10], Anne Lærke Spangmose[7], Christina Bergh[1☯], Ulla-Britt Wennerholm[1☯]*

**1** Department of Obstetrics and Gynecology, Institute of Clinical Sciences, Sahlgrenska Academy, University of Gothenburg, Sahlgrenska University Hospital, Gothenburg, Sweden, **2** Department of Pediatrics, Institute of Clinical Sciences, Sahlgrenska Academy, University of Gothenburg, Sahlgrenska University Hospital, Gothenburg, Sweden, **3** School of Public Health and Community Medicine, Institute of Medicine, University of Gothenburg, Gothenburg, Sweden, **4** Department of Public Health and Nursing, Norwegian University of Science and Technology, Trondheim, Norway, **5** THL Finnish Institute for Health and Welfare, Information Services Department, Helsinki, Finland, **6** Karolinska Institute, Department of Molecular Medicine and Surgery, Stockholm, Sweden and Region Stockholm, Academic Primary Health Care Center, Stockholm, Sweden, **7** The Fertility Clinic, Copenhagen University Hospital, Rigshospitalet, Copenhagen, Denmark, **8** Department of Obstetrics and Gynecology, Helsinki University Hospital and University of Helsinki, Finland, **9** Center for Fertility and Health, Norwegian Institute of Public Health, Oslo, Norway, **10** Spiren Fertility Clinic, Trondheim, Norway

☯ These authors contributed equally to this work.
* ulla-britt.wennerholm@vgregion.se

**Data Availability Statement:** Data cannot be shared publicly owing to restrictions by law. Data are stored at CoNARTaS folder at Statistics Denmark's server, after receiving approvals by the

## Abstract

### Background

The aim was to investigate whether children born after assisted reproduction technology (ART), particularly after frozen-thawed embryo transfer (FET), are at higher risk of childhood cancer than children born after fresh embryo transfer and spontaneous conception.

### Methods and findings

We performed a registry-based cohort study using data from the 4 Nordic countries: Denmark, Finland, Norway, and Sweden. The study included 7,944,248 children, out of whom 171,774 children were born after use of ART (2.2%) and 7,772,474 children were born after spontaneous conception, representing all children born between the years 1994 to 2014 in Denmark, 1990 to 2014 in Finland, 1984 to 2015 in Norway, and 1985 to 2015 in Sweden. Rates for any cancer and specific cancer groups in children born after each conception method were determined by cross-linking national ART registry data with national cancer and health data registries and population registries. We used Cox proportional hazards models to estimate the risk of any cancer, with age as the time scale.

After a mean follow-up of 9.9 and 12.5 years, the incidence rate (IR) of cancer before age 18 years was 19.3/100,000 person-years for children born after ART (329 cases) and 16.7/100,000 person-years for children born after spontaneous conception (16,184 cases). Adjusted hazard ratio (aHR) was 1.08, 95% confidence interval (CI) 0.96 to 1.21, $p$ = 0.18. Adjustment was performed for sex, plurality, year of birth, country of birth, maternal age at

Ethics Committees and registry keeping authorities in each country, as described in the following publication: Opdahl S, Henningsen AA, Bergh C, Gissler M, Romundstad LB, Petzold M, Tiitinen A, Wennerholm UB, Pinborg AB. Data Resource Profile: Committee of Nordic Assisted Reproductive Technology and Safety (CoNARTaS) cohort. Int J Epidemiol. 2020 Apr 1;49(2):365-366f. doi: 10.1093/ije/dyz228. Contact information for Statistics Denmark: Division of Research Services Statistics Denmark Sejrøgade 11 DK-2100 Copenhagen Denmark E-mail: forskningsservice@dst.dk Phone: +45 39 17 31 30.

**Funding:** The CoNARTaS has been supported by the Nordic Trial Alliance: a pilot project jointly funded by the Nordic Council of Ministers and NordForsk [grant number 71450] (AP), the Central Norway Regional Health Authorities [grant number 46045000] (SO), the Norwegian Cancer Society [grant number 182356–2016] SO), the Nordic Federation of Obstetrics and Gynaecology [grant numbers NF13041, NF15058, NF16026 and NF17043] (UBW, AT), the Interreg Öresund-Kattegat-Skagerrak European Regional Development Fund (ReproUnion project) (AP), and by the Research Council of Norway's Centre of Excellence funding scheme [grant number 262700] (SO), the Swedish state under the agreement between the Swedish government and the county councils, the ALF-agreement (ALFGBG-70940) (CB), the Hjalmar Svensson Foundation (UBW), and The Swedish Childhood Cancer Foundation (BL). The funding sources had no role in study design; in the collection, analysis, and interpretation of data; in the writing of the report; and in the decision to submit the article for publication.

**Competing interests:** The authors have declared that no competing interests exist.

**Abbreviations:** aHR, adjusted hazard ratio; ART, assisted reproduction technology; CI, confidence interval; CNS, central nervous system; FET, frozen-thawed embryo transfer; ICCC-3, International Classification of Childhood Cancer; ICD, International Statistical Classification of Diseases; IR, incidence rate; WHO, World Health Organization.

birth, and parity. Children born after FET had a higher risk of cancer (48 cases; IR 30.1/100,000 person-years) compared to both fresh embryo transfer (IR 18.8/100,000 person-years), aHR 1.59, 95% CI 1.15 to 2.20, $p = 0.005$, and spontaneous conception, aHR 1.65, 95% CI 1.24 to 2.19, $p = 0.001$. Adjustment either for macrosomia, birth weight, or major birth defects attenuated the association marginally. Higher risks of epithelial tumors and melanoma after any assisted reproductive method and of leukemia after FET were observed.

The main limitation of this study is the small number of children with cancer in the FET group.

## Conclusions

Children born after FET had a higher risk of childhood cancer than children born after fresh embryo transfer and spontaneous conception. The results should be interpreted cautiously based on the small number of children with cancer, but the findings raise concerns considering the increasing use of FET, in particular freeze-all strategies without clear medical indications.

## Trial registration

Trial registration number: ISRCTN 11780826.

## Author summary

### Why was this study done?

- Worldwide, the number of children born after assisted reproductive technology (ART) with frozen-thawed embryo transfer (FET) increases and now exceeds the number of children born after fresh embryo transfer in many countries.

- Singletons born after FET are at increased risk of macrosomia that has been associated with a higher risk of childhood cancer.

- Studies on the association of ART and risk of childhood cancer show conflicting results.

### What did the researchers do and find?

- We performed a Nordic registry-based cohort study including 171,774 children born after use of ART and 7,772,474 children born after spontaneous conception.

- We found that children born after FET had a higher risk of childhood cancer than children born after fresh embryo transfer and spontaneous conception. We found no increase in childhood cancer after any ART.

### What do these findings mean?

- Concerns may be raised considering the vast increase in FET, in particular freeze-all strategies without clear medical indications.

- The main limitation of this study is the small number of children with cancer in the FET group.

## Introduction

Recently, a substantial increase in use of frozen-thawed embryo transfers (FETs) in in vitro fertilization has occurred worldwide. In the United States of America, the FET rate has doubled since 2015 and comprised 78.8% of all embryo transfers using non-donor assisted reproductive technology (ART) in 2019 [1]. A similar pattern is observed in Australia, New Zealand, and Europe [2]. The main reason for the increase in FET is improved embryo survival and the high pregnancy/live birth rates after transfer of vitrified/thawed blastocysts compared to the previously used technique with transfer of slow frozen-thawed cleavage stage embryos [3,4]. A freeze-all policy (freezing of all embryos from a treatment cycle and no fresh embryo transfer) is currently being implemented in many parts of the world [2], despite indications of increased birth weight and risk of hypertensive disorders in pregnancy [5] and without careful consideration of benefits and harms. Six large randomized controlled trials have investigated the differences in live birth rate following fresh embryo transfer and FET in freeze-all cycles [6–11]. The first trial, published in 2016 [6], showed a significantly higher live birth rate in freeze-all groups than fresh embryo transfer groups in anovulatory women. In ovulatory women, most trials show similar ongoing pregnancy and live birth rates in a freeze-all group (either cleavage stage embryos or blastocysts) compared with a fresh embryo transfer group [7,8,10,11]. Importantly, freezing has reduced multiple pregnancies by facilitating single embryo transfer [12], and the freeze-all strategy has almost eliminated ovarian hyperstimulation syndrome [5,13], a potentially life-threatening complication in ART [14]. Currently, up to 7.9% of children in Europe and 5.1% in the United States are born after ART, making health of children born after ART a topic of public health importance [15,16].

In many countries, the number of FET-conceived children has now exceeded the number born after fresh embryo transfer [1,17].

Childhood cancer includes a wide array of diagnoses, some of them very rare. Often the diagnoses are seen only in children but also cancer diseases common in adults occur. Leukemia is the most common neoplasm followed by various forms of tumors in the central nervous system (CNS). The incidence peaks during the first years of life [18]. The overall incidence in Northern Europe increased slightly up to the turning of the century, but later on, a stabilization has followed [19]. Studies on risk of childhood cancer after ART show conflicting results. Most large observational studies indicate similar overall cancer risk in children born after ART and in children in the general population [20–23], but a higher risk for both any cancer [24–27] and specific malignancies [20,21,24–26] has also been reported. In a Danish population-based registry study [22], a higher risk of any childhood cancer was found after FET compared to spontaneous conception, but the finding was based on a limited number of cases.

In this large population-based registry study from 4 Nordic countries, we estimated the risk of childhood cancer in an unselected ART-conceived population, with special focus on children born after FET, and compared it to the risk in children born after fresh embryo transfer and spontaneous conception during the same period.

## Methods

### Study population and data collection

Data were obtained for Denmark, Finland, Norway, and Sweden from the CoNARTaS (Committee of Nordic ART and Safety) cohort [28], established to study short- and long-term health consequences of ART treatment in children and their mothers. Data on maternal and perinatal health in all deliveries were obtained from nationwide Medical Birth Registries in each country [29] and cross-linked with data from the national cancer registries, national patient registries, the national cause of death registries, and socioeconomic data retrieved from the population registries in each country. The unique personal identity number assigned to each resident in the Nordic countries enabled individual-level data linkage between registries and between children and their mothers.

All Nordic cancer registries are population based and nationwide. The respective cancer registries were founded in 1942 in Denmark [30], 1952 in Finland [31], 1951 in Norway [32], and 1958 in Sweden [33]. Notification of cancer is mandatory in all Nordic countries. A high degree of completeness and accuracy of the registered data and comparability between countries has been documented [34].

ART conception was determined from reports to the Medical Birth Registry (Finland), notifications from fertility clinics regarding all ongoing ART-conceived pregnancies in gestational weeks 6 to 7 (Norway) or the National Board of Health and Welfare (Sweden until 2007), or through linkage with cycle-based ART registries (Denmark, Sweden from 2007) (S1 Table). Details on the cohort and registries are given elsewhere [28].

Inclusion criteria were all live-born singletons, twins, and higher order multiples born after ART and spontaneous conception (here defined as any conception without ART) during the study period.

### Outcome variables and follow-up

The primary outcomes were any cancer diagnosed before age 18 years after use of any ART and specifically after FET. Secondary outcomes were cancer diagnosis groups according to the International Classification of Childhood Cancer (ICCC-3) [35]. The ICCC-3 is based on the World Health Organization (WHO) classification system for cancer morphology and allows comparison of broad categories of neoplasms in continuity with previous classifications [36]. In ICCC-3, the diagnoses are grouped into 12 main categories according to the morphology code, the topographic code, and the behavior of the tumor, i.e., benign or malignant (S2 Table). We grouped all patients into ICCC-3 categories. The 12 groups each include a defined set of morphology codes, and occasionally, the additional use of topography codes was used. In older patients topography codes according to the International Statistical Classification of Diseases (ICD) and Related Health problems were transferred to the latest version, ICD-10, by an algorithm used by the cancer registries. ICCC-3 only groups tumors with a malignant diagnosis except for tumors located in the CNS. Consequently, other benign or borderline tumors were not included in this report. Although there are discrepancies, due mainly to different traditions in cancer registration between the countries [28,34], pooling of data was possible because all use the WHO classification system [36].

Macrosomia was defined as birth weight ≥4,000 g. Birth defects and chromosomal aberrations were defined according to ICD-9 (740–759) or ICD-10 (Q00–99) code. Major birth defects were defined according to the EUROCAT classification system (S1 Table) [37].

This study is reported as per the Strengthening the Reporting of Observational Studies in Epidemiology (STROBE) guideline (S1 STROBE Checklist). Our analyses were planned in

advance of the research team accessing any data, and our study protocol is provided (S1 Text). The CoNARTaS project is also registered in the ISRCTN registry (ISRCTN11780826).

## Ethical approval

Ethical approval was obtained from Ethical Committee in Gothenburg, Sweden (Dnr 214–12, T422-12, T516-15, T233-16, T300-17, T1144-17, T121-18, T1071-18, 2019–02347). In Norway, approval was given by the Regional Committee for Medical and Health (REK-Nord, 2010/1909). There are no requirements for ethical approval for registry-based studies in Denmark and Finland. All registry-keeping organizations gave permission to use their data in this study.

## Statistical analysis

We used Cox proportional hazards models to estimate the risk of any cancer, with age as the time scale. We computed each child's time at risk from date of birth until whichever event occurred first: diagnosis of any cancer, emigration (available through 2014 for Denmark, through 2015 for Sweden and Norway, and not available for Finland), death (available through 2014 for Denmark and Finland and 2015 for Norway and Sweden), 18th birthday, or end of the follow-up period (December 31, 2014 for Finland, December 31, 2015 for Norway and Sweden, and December 31, 2018 for Denmark).

We compared risk of cancer between children born after ART and spontaneous conception, between children born after FET and fresh embryo transfer, and between children born after FET and spontaneous conception, for any cancer and the 12 different cancer groups. In all analyses, only the first diagnosed cancer type was considered. Finland was not included in the analysis of FET since the Finnish registration does not differentiate between different assisted reproduction methods. We further analyzed risk of any cancer for singletons and multiples separately.

We estimated crude and adjusted hazard ratios (HRs) with 95% confidence intervals (CIs). The significance level was set to 5%. A number of <10 events in any group was considered too small to calculate a stable estimate. Adjustments were made for selected covariates. Selection of covariates was primarily based on medical knowledge and previous studies. We searched literature for identification of covariates [38]. The variables included the following child and maternal characteristics: calendar year of birth (continuous variable), country of birth (Denmark, Finland, Norway, Sweden), maternal age at delivery (continuous variable), parity (nulliparous/parous), sex, and plurality (singletons/multiples). Calendar year at birth and country of birth both influence cancer incidence as well as likeliness of having been conceived by ART. Both maternal age and birth order have been shown to be associated with cancer in offspring [39,40] and are also associated with ART (ART mothers are older and of lower parity than spontaneous conception mothers). Risk of certain cancers is different among males and females [41], and some ART methods (transfer of blastocysts) may alter the sex ratio [42], and therefore, sex was included as a covariate. Furthermore, an association with multiple birth and cancer (leukemia) has been found [43] and multiple birth is more common after ART, and plurality was therefore also included as a covariate.

In a sensitivity analysis, we also included maternal smoking during pregnancy (yes/no) as a covariate. In an additional sensitivity analysis, maternal highest educational level achieved during the study period (low, medium, high) was included as a covariate [44]. This analysis included data from Denmark, Finland, and Sweden because data on education were not available from Norway.

In the main regression analysis where adjustment was performed for year of birth, country of birth, maternal age at birth, parity, sex, and plurality, the percentage of missing data was

small. In the sensitivity analyses where adjustment was performed for maternal smoking or educational level, missing data for these variables were substantial. Participants with missing data were excluded from these models. No imputations were made.

Macrosomia and major birth defects have been associated with childhood cancer [45–47] and are also associated with ART [23,48]. To investigate macrosomia and major birth defects, as possible mechanisms of an increased risk of cancer in children born after FET, separate exploratory analyses were performed with additional adjustment for macrosomia (yes/no) and major birth defects (yes/no). A similar analysis was also performed with birth weight as a continuous variable. Finally, as an indicator of embryo quality, we additionally adjusted for embryo stage, i.e., cleavage stage or blastocyst in a separate exploratory analysis comparing conception after FET and fresh embryo transfer.

Collinearity was assessed via the post-estimation command estat variance–covariance matrix of the estimators (VCE) in Stata, giving the covariances/correlations between the different covariates in the Cox proportional hazards model. No major issues with multicollinearity were identified in our analyses.

The proportional hazards assumption was tested with Schoenfeld residuals, and there were no clear violations. All analyses were performed in Stata, version 16.

### Patient and public involvement

Children or parents were not involved in the design, outcome measures, or planning of the study, and they were not asked to give advice on interpretation of results. The results of the research will be disseminated to the public through broadcasts, popular science articles, and newspapers.

## Results

### Child and maternal characteristics

The study population included 171,774 children born after use of ART and 7,772,474 children born after spontaneous conception (S1 Fig). Child and maternal characteristics are presented in Table 1 for any ART method and spontaneous conception and in S3 Table for FET, fresh embryo transfer, and spontaneous conception. Overall, 25.9% and 2.6% of the children born after ART and spontaneous conception were part of a multifetal pregnancy. Preterm birth (<37 weeks) and low birth weight (<2,500 g) occurred among 16.1% and 13.0% of the children born after ART and among 5.6% and 3.5% of children born after spontaneous conception. Mean maternal age at delivery was 33.9 and 29.7 years in the ART and spontaneously conceived population, and 68.1% and 41.8% of the mothers were primiparous.

### Risk of cancer after ART-conception

The total follow-up time was 1,705,772 person-years for the ART group (mean [SD] 9.9 [5.8] years) and 97,027,051 person-years for the spontaneously conceived group (mean [SD] 12.5 [5.9] years). Cancer was diagnosed before age 18 years in 329 children in the ART group (incidence rate (IR) 19.3 per 100,000 person-years, Table 2) and in 16,183 spontaneously conceived children (IR 16.7/100,000 person-years). The mean age at first cancer diagnosis was 6.0 years after ART and 6.8 years after spontaneous conception, and the distribution of age at first cancer diagnosis (Fig 1) reflected the longer mean follow-up after spontaneous conception. Age-specific hazard rates were slightly higher among ART-conceived compared to spontaneously conceived children from approximately 5 to 12 years of age (Fig 2), corresponding to unadjusted cumulative hazards that were similar up to about 6 years of age and diverged slightly

**Table 1. Characteristics of study population by mode of conception defined as ART or SC and by country of birth in children born in Denmark 1994–2014, Finland 1990–2014, Norway 1984–2015, or Sweden 1985–2015.**

| | All countries N = 7,944,248 | | Denmark N = 1,355,267 | | Finland N = 1,496,133 | | Norway N = 1,865,484 | | Sweden N = 3,227,364 | |
|---|---|---|---|---|---|---|---|---|---|---|
| | ART N = 171,774 | SC N = 7,772,474 | ART N = 45,783 | SC N = 1 309,484 | ART N = 29,682 | SC N = 1,466,451 | ART N = 34,042 | SC N = 1,831,442 | ART N = 62,267 | SC N = 3,165,097 |
| Child characteristics | | | | | | | | | | |
| *Calendar year of birth, N (%)* | | | | | | | | | | |
| 1984–1990 | 1,676 (1.0) | 1,095,853 (14.1) | - | - | 53 (0.2) | 65,203 (4.5) | 877 (2.6) | 383,757 (21.0) | 746 (1.2) | 646,893 (20.4) |
| 1991–1995 | 11,681 (6.8) | 1,321,054 (17.0) | 1,299 (2.8) | 138,138 (10.6) | 2,864 (9.7) | 320,916 (21.9) | 2,464 (7.2) | 297,356 (16.2) | 5,054 (8.1) | 564,644 (17.8) |
| 1996–2000 | 28,705 (16.7) | 1,133,572 (17.2) | 8,532 (18.6) | 327,305 (25.0) | 6,937 (23.4) | 283,142 (19.3) | 4,313 (12.7) | 292,015 (15.9) | 8,923 (14.3) | 431,110 (13.6) |
| 2001–2005 | 36,089 (21.0) | 1,332,301 (17.1) | 11,797 (25.8) | 313,095 (23.9) | 6,357 (21.4) | 276,286 (18.8) | 6,586 (19.4) | 276,926 (15.1) | 11,349 (18.2) | 465,994 (14.7) |
| 2006–2010 | 45,499 (26.5) | 1,414,136 (18.2) | 13,139 (28.7) | 310,160 (23.7) | 6,971 (23.5) | 291,843 (19.9) | 9,411 (27.7) | 293,014 (16.0) | 15,978 (25.7) | 519,119 (16.4) |
| 2011–2015 | 48,124 (28.0) | 1,275,558 (16.4) | 11,016 (24.1) | 220,786 (16.9) | 6,500 (21.9) | 229,061 (15.6) | 10,391 (30.5) | 288,374 (15.8) | 20,217 (32.5) | 537,337 (17.0) |
| *Birth weight, N (%)* | | | | | | | | | | |
| Very low birth weight, <1,500 g | 5,220 (3.1) | 56,245 (0.7) | 1,566 (3.5) | 10,331 (0.8) | 844 (2.8) | 9,494 (0.7) | 1,233 (3.6) | 14,584 (0.8) | 1,577 (2.5) | 21,836 (0.7) |
| Low birth weight, <2,500 g | 22,241 (13.0) | 272,089 (3.5) | 6,894 (15.2) | 50,901 (4.0) | 3,936 (13.3) | 46,765 (3.2) | 4,870 (14.3) | 67,004 (3.7) | 6,541 (10.6) | 107,419 (3.4) |
| **Macrosomia,** ≥4,000 g | 20,522 (12.0) | 1,458,901 (18.9) | 4,723 (10.4) | 238,105 (18.5) | 3,521 (11.9) | 272,308 (18.6) | 3,933 (11.6) | 355,748 (19.4) | 8,345 (13.5) | 592,740 (18.8) |
| Birth weight, g, mean (SD) | 3,193 (754) | 3,517 (584) | 3,109 (764) | 3,492 (596) | 3,200 (745) | 3,527 (566) | 3,155 (780) | 3,522 (595) | 3,272 (729) | 3,520 (582) |
| Missing data for birth weight, N (%) | 755 (0.4) | 36,522 (0.5) | 463 (1.0) | 24,570 (1.9) | 13 (0.04) | 3,405 (0.2) | 35 (0.1) | 1,556 (0.1) | 244 (0.4) | 6,991 (0.2) |
| *Gestational age, N (%)* | | | | | | | | | | |
| Extremely preterm birth, <28+0 weeks | 2,100 (1.2) | 22,136 (0.3) | 677 (1.5) | 4,075 (0.3) | 341 (1.2) | 3,800 (0.3) | 476 (1.4) | 5,627 (0.3) | 606 (1.0) | 8,634 (0.3) |
| Very preterm birth, <32+0 weeks | 4,364 (2.6) | 47,876 (0.6) | 1,334 (2.9) | 9,355 (0.7) | 681 (2.3) | 7,616 (0.5) | 978 (2.9) | 12,110 (0.7) | 1,371 (2.2) | 18,795 (0.6) |
| Preterm birth, <37+0 weeks | 27,462 (16.1) | 428,385 (5.6) | 8,364 (18.4) | 82,501 (6.4) | 5,125 (17.3) | 74,474 (5.1) | 5,842 (17.3) | 99,294 (5.7) | 8,131 (13.1) | 172,116 (5.5) |
| Postterm birth, ≥42+0 weeks | 5,057 (3.0) | 374,569 (4.9) | 789 (1.7) | 40,354 (3.1) | 573 (1.9) | 40,738 (2.8) | 1,152 (3.4) | 124,498 (7.2) | 2,543 (4.1) | 168,979 (5.4) |
| Gestational age, days, mean (SD) | 270 (20) | 278 (14) | 268 (21) | 278 (14) | 270 (20) | 278 (13) | 270 (21) | 279 (14) | 273 (19) | 278 (13) |
| Missing data for gestational age, N (%) | 652 (0.4) | 129,426 (1.7) | 298 (0.7) | 28,949 (2.2) | 57 (0.2) | 6,738 (0.5) | 246 (0.7) | 89,599 (4.9) | 51 (0.1) | 4,140 (0.1) |
| *Plurality, N (%)* | | | | | | | | | | |
| Singletons | 127,230 (74.1) | 7,573,456 (97.4) | 30,997 (67.7) | 1,269,481 (97.0) | 22,038 (74.3) | 1,430,582 (97.6) | 24,114 (70.1) | 1,783,892 (97.4) | 50,081 (80.4) | 3,089,501 (97.6) |
| Twins | 42,536 (24.8) | 194,464 (2.5) | 14,392 (31.4) | 38,981 (3.0) | 7,196 (24.2) | 35,165 (2.4) | 9,367 (27.5) | 46,389 (2.5) | 11,581 (18.6) | 73,929 (2.3) |

*(Continued)*

**Table 1.** (Continued)

| | All countries N = 7,944,248 | | Denmark N = 1,355,267 | | Finland N = 1,496,133 | | Norway N = 1,865,484 | | Sweden N = 3,227,364 | |
|---|---|---|---|---|---|---|---|---|---|---|
| | ART N = 171,774 | SC N = 7,772,474 | ART N = 45,783 | SC N = 1 309,484 | ART N = 29,682 | SC N = 1,466,451 | ART N = 34,042 | SC N = 1,831,442 | ART N = 62,267 | SC N = 3,165,097 |
| Triplets and higher order multiples | 2,008 (1,2) | 4,554 (0.1) | 394 (0.9) | 1,022 (0.1) | 448 (1.5) | 704 (0.1) | 561 (1.7) | 1,161 (0.1) | 605 (1.0) | 1,667 (0.1) |
| *Birth defects*, N (%) | | | | | | | | | | |
| Any major defects[a] (non-chromosomal or chromosomal defects) | 8,965 (5.2) | 263,781 (3.4) | 2,597 (5.6) | 48,686 (3.7) | 2,165 (7.3) | 67,234 (4.6) | 1,802 (5.3) | 62,089 (3.4) | 2,401 (3.8) | 85,772 (2.7) |
| Major birth defects[a] (non-chromosomal) | 8,679 (5.1) | 256,525 (3.3) | 2,536 (5.5) | 47,388 (3.6) | 2,099 (7.1) | 65,291 (4.5) | 1,733 (5.1) | 60,506 (3.3) | 2,311 (3.7) | 83,340 (2.6) |
| Chromosomal defects (with or without other major birth defects[a]) | 286 (0.17) | 7,256 (0.09) | 61 (0.13) | 1,298 (0.10) | 66 (0.22) | 1,943 (0.13) | 69 (0.20) | 1,583 (0.09) | 90 (0.14) | 2,432 (0.08) |
| Male sex, N (%) | 87,805 (51.1) | 3,988,987 (51.3) | 23,202 (50.7) | 672,088 (51.3) | 15,192 (51.2) | 749,359 (51.1) | 17,456 (51.3) | 940,571 (51.4) | 31,955 (51.3) | 1,626,969 (51.4) |
| Age at cancer diagnosis (year), mean (SD), median (range) | 6.0 (5.0) 4.3 (0–18) | 6.8 (5.4) 5.2 (0–18) | 7.0 (5.3) 5.2 (0–17) | 6.9 (5.4) 5.3 (0–18) | 5.9 (4.8) 4.7 (0–17) | 6.6 (5.3) 4.8 (0–18) | 5.4 (4.8) 4.1 (0–18) | 7.3 (5.6) 5.7 (0–18) | 5.2 (4.5) 3.5 (0–17) | 6.7 (5.3) 5.0 (0–18) |
| Follow-up time (year), mean (SD), median (range) | 9.9 (5.8) 9.5 (0–18) | 12.5 (5.9) 14.5 (0–18) | 12.0 (4.8) 12.2 (0–18) | 13.2 (4.9) 14.4 (0–18) | 10.0 (5.9) 10.1 (0–18) | 11.7 (6.0) 12.8 (0–18) | 9.0 (5.8) 8.2 (0–18) | 12.8 (6.0) 15.7 (0–18) | 8.9 (5.9) 7.9 (0–18) | 12.4 (6.2) 15.0 (0–18) |
| Maternal characteristics | | | | | | | | | | |
| Age at delivery, (year), mean (SD) | 33.9 (4.2) | 29.7 (5.2) | 33.6 (4.1) | 30.2 (4.9) | 33.9 (4.6) | 29.8 (5.3) | 33.6 (4.1) | 29.3 (5.2) | 34.2 (4.1) | 29.8 (5.2) |
| Primiparous, N (%) | 116,551 (68.1) | 3,244,158 (41.8) | 30,479 (67.4) | 554,294 (42.9) | 20,132 (67.9) | 592,796 (40.5) | 21,733 (63.8) | 756,578 (41.3) | 44,207 (71.0) | 1,340,490 (42.4) |
| Smoking during pregnancy[b], N (%) | 10,141 (6.4) | 989,017 (15.1) | 3,956 (9.3) | 194,512 (16.5) | 1,820 (6.2) | 224,725 (15.7) | 1,735 (6.1) | 129,171 (13.4) | 2,630 (4.5) | 440,609 (14.7) |
| Missing data for smoking, N (%) | 12,944 (7.5) | 1,213,569 (15.6) | 3,217 (7.0) | 132,470 (10.1) | 447 (1.5) | 37,222 (2.5) | 5,703 (16.8) | 868,891 (47.4) | 3,577 (5.7) | 174,986 (5.5) |
| BMI (kg/m²), mean (SD) | 24.3 (4.1) | 24.2 (4.5) | 24.0 (4.3) | 24.3 (5.0) | 24.1 (4.3) | 24.3 (4.8) | 24.4 (4.4) | 24.3 (4.8) | 24.4 (3.9) | 24.1 (4.3) |
| Missing data for BMI, N (%) | 64,265 (37.4) | 3,814,698 (49.1) | 17,836 (40.0) | 693,037 (52.9) | 14,610 (49.2) | 866,154 (59.1) | 24,402 (71.7) | 1,575,159 (86.0) | 7,417 (11.9) | 680,348 (21.5) |
| *Educational level, N (%)[c, d]* | | | | | | | | | | |
| Low (ISCED <5) | 59,198 (44.7) | 3,103,182 (56.1) | 24,091 (53.3) | 775,825 (61.0) | 8,674 (30.9) | 604,310 (46.2) | NA | NA | 26,433 (44.7) | 1,723,047 (58.3) |
| Medium (ISCED 5–6) | 45,322 (34.2) | 1,651,780 (29.8) | 13,657 (30.1) | 342,858 (26.9) | 11,349 (40.4) | 466,435 (35.7) | NA | NA | 20,316 (34.3) | 842,487 (28.5) |
| High (ISCED 7–8) | 27,952 (21.1) | 781,679 (14.1) | 7,455 (16.5) | 154,279 (12.1) | 8,078 (28.8) | 237,128 (18.1) | NA | NA | 12,419 (20.1) | 390,272 (13.2) |
| Missing data for educational level | 5,260 (4.0) | 404,391 (7.3) | 580 (1.3) | 36,522 (2.8) | 1,581 (5.3) | 158,578 (10.8) | NA | NA | 3,099 (5.0) | 209,291 (6.6) |
| *Assisted reproduction method[e], N (%)* | | | | | | | | | | |

(Continued)

**Table 1.** (Continued)

| | All countries N = 7,944,248 | | Denmark N = 1,355,267 | | Finland N = 1,496,133 | | Norway N = 1,865,484 | | Sweden N = 3,227,364 | |
|---|---|---|---|---|---|---|---|---|---|---|
| | ART N = 171,774 | SC N = 7,772,474 | ART N = 45,783 | SC N = 1 309,484 | ART N = 29,682 | SC N = 1,466,451 | ART N = 34,042 | SC N = 1,831,442 | ART N = 62,267 | SC N = 3,165,097 |
| IVF | 81,948 (57.7) | - | 25,006 (54.6) | - | NA | - | 20,093 (59.0) | - | 36,849 (59.2) | - |
| ICSI | 55,126 (38.8) | - | 18,118 (39.6) | - | NA | - | 11,590 (34.0) | - | 25,418 (40.8) | - |
| Missing data for IVF/ICSI | 5,018 (3.5) | - | 2,659 (5.8) | - | NA | - | 2,359 (7.0) | - | 0 | - |
| Fresh embryo transfer | 115,474 (81.3) | - | 41,022 (89.6) | - | NA | - | 25,630 (75.3) | - | 48,822 (78.4) | - |
| Frozen embryo transfer | 22,630 (15.9) | - | 4,761 (10.4) | - | NA | - | 4,424 (13.0) | - | 13,445 (21.6) | - |
| Missing data for fresh/frozen embryo transfer | 3,988 (2.8) | - | 0 | - | NA | - | 3,988 (11.7) | - | 0 | - |
| Cleavage stage embryo | 130,784 (92.0) | - | 42,444 (92.7) | - | NA | - | 33,628 (98.8) | - | 54,712 (87.9) | - |
| Blastocysts | 9,623 (6.8) | | 1,654 (3.6) | | | | 414 (1.2) | | 7,555 (12.1) | |
| Missing data for embryo stage | 1,685 (1.2) | - | 1,685 (3.7) | - | NA | - | 0 | - | 0 | - |
| Autologous oocytes | 140,682 (99.0) | - | 45,073 (98.4) | - | NA | - | 34,042 (100) | - | 61,567 (98.9) | - |
| Donated oocytes | 1,410 (1.0) | - | 710 (1.6) | - | NA | - | 0[f] | - | 700 (1.1) | - |

[a]Major birth defects defined according to the EUROCAT classification system [37].

[b]Data for Denmark, Finland, and Sweden but only birth cohorts since 1999 from Norway when smoking habits were first registered.

[c]Data for Denmark, Finland, and Sweden because no data were available for Norway.

[d]Educational level according to International Standard Classification of Education (ISCED2011), ISCED <5 = primary, secondary, or post-secondary non tertiary education, ISCED 5–6 = first stage of tertiary education (bachelors or equivalent), ISCED 7–8 = second stage of tertiary education (master, doctorate, or more) [44].

[e]Data for Denmark, Norway, and Sweden because no data on assisted reproductive method were available for Finland.

[f]Oocyte donation not permitted in Norway.

ART, assisted reproductive technology; BMI, body mass index; ICSI, intracytoplasmic sperm injection; ISCED, international standard classification of education; IVF, in vitro fertilization; LGA, large for gestational age; NA, not available; SC, spontaneous conception; SGA, small for gestational age.

thereafter (Fig 3). After adjustment, no statistically significant difference in any cancer risk was found for children born after ART versus spontaneous conception (aHR 1.08, 95% CI 0.96 to 1.21, $p = 0.18$) (Table 3).

The 2 most common cancer types were leukemia and CNS tumors (Table 3). There were 111 cases of leukemia among children born after ART (IR 6.5/100,000 person-years) and 4,921 cases after spontaneous conception (IR 5.1/100,000 person-years) (aHR 1.09, 95% CI 0.89 to 1.33, $p = 0.40$). The rates of any chromosomal aberration among children with leukemia were 4.5% in ART and 2.2% in the spontaneous conception group.

CNS tumors occurred in 87 children born after ART and 4,080 after spontaneous conception (IR 5.1 and 4.2/100,000 person-years, respectively) (aHR 1.22, 95% CI 0.97 to 1.52, $p = 0.09$). A higher risk of epithelial tumors and melanoma was found in children born after ART (22 cases, IR 1.3/100,000 person-years) compared with children born after spontaneous conception (812 cases, IR 0.8/100,000 person-years) (aHR 1.89, 95% CI 1.20 to 2.97, $p = 0.01$).

**Table 2. IR of any cancer before 18 years of age by mode of conception and country of birth in children born in Denmark, Finland, Norway, or Sweden (Denmark 1994–2014, Finland 1990–2014, Norway 1984–2015, and Sweden 1985–2015).**

| | ART | | | Spontaneous conception | | | All | | |
|---|---|---|---|---|---|---|---|---|---|
| | No. of children with cancer | IR | | No. of children with cancer | IR | | No. of children with cancer | IR | |
| | | Per 1,000 children | Per 100,000 person-years | | Per 1,000 children | Per 100,000 person-years | | Per 1,000 children | Per 100,000 person-years |
| **All countries** | 329 | 1.92 (329/ 171,774) | 19.29 (329/1,705,772) | 16,184 | 2.08 (16,184/ 7,772,474) | 16.68 (16 184/ 97,027,051) | 16,513 | 2.08 (16,513/ 7,944,248) | 16.72 (16,513/ 98,732,823) |
| **Denmark** | 108 | 2.36 (108/ 45,783) | 19.29 (108/549,372) | 2,840 | 2.17 (2,840/ 1,309,484) | 16.48 (2,840/ 17,231,434) | 2,948 | 2.18 (2,948/ 1,355,267) | 16.58 (2,948/ 17,780,806) |
| **Finland** | 49 | 1.65 (49/29,682) | 19.66 (49/296,659) | 3,140 | 2.14 (3,140/ 1,466,451) | 18.36 (3,140/ 17,106,671) | 3,189 | 2.13 (3,189/ 1,496,133) | 18.32 (3,189/ 17,403,330) |
| **Norway** | 63 | 1.85 (63/34,042) | 16.52 (63/306,537) | 3,841 | 2.10 (3,841/ 1,831,442) | 16.43 (3,841/ 23,373,870) | 3,904 | 2.09 (3,904/ 1,865,484) | 16.49 (3,904/ 23,680,407) |
| **Sweden** | 109 | 1.75 (109/ 62,267) | 20.55 (109/553,204 | 6,363 | 2.01 (6,363/ 3,165,097) | 16.18 (6,363/ 39,315,076) | 6,472 | 2.01 (6,472/ 3,227,364) | 16.23 (6,472/ 39,868,280) |

ART, assisted reproduction technology; IR, incidence rate.

No significant differences were observed for other types of cancer where statistical comparisons were performed.

The IRs for any cancer and different cancer types by country of birth are presented in S4 Table.

Sensitivity analyses, including adjustment for any smoking during pregnancy or highest maternal educational level, only marginally changed the association (aHR 1.02, 95% CI 0.90 to 1.15, $p = 0.75$ and aHR 1.08, 95% CI 0.95 to 1.22, $p = 0.27$, respectively).

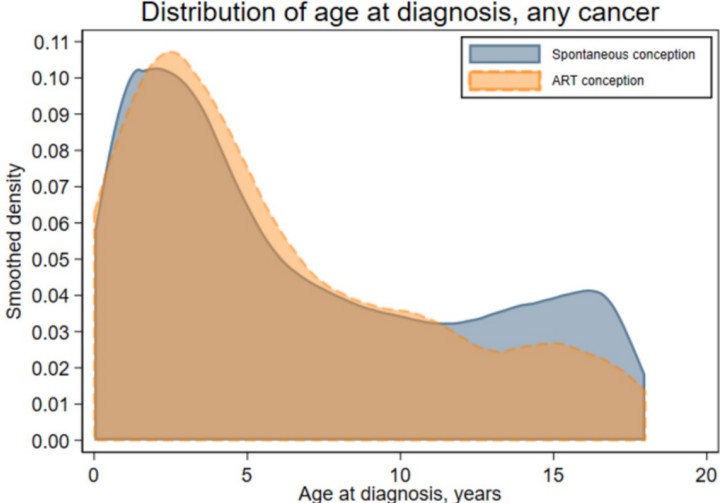

**Fig 1. Proportional distribution of age at first cancer (any type) among spontaneously and ART-conceived children born in Denmark (1994–2014), Finland (1990–2014), Norway (1984–2015), and Sweden (1985–2015) and diagnosed with cancer before age 18 years.** ART, assisted reproduction technology.

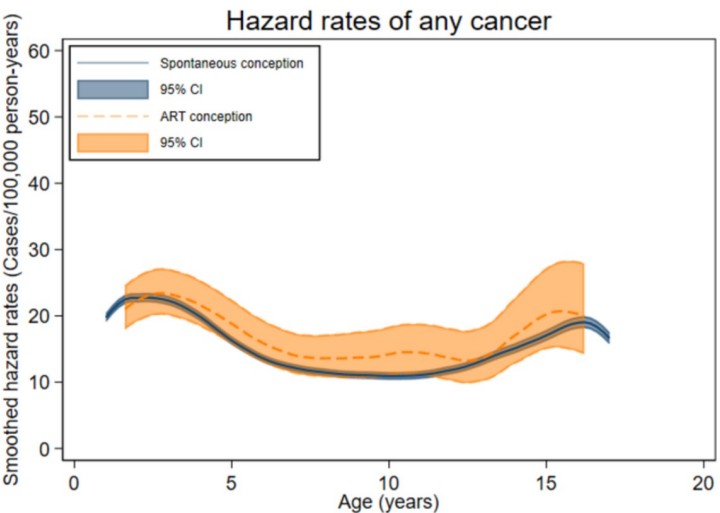

**Fig 2. Age-specific hazard rates of first cancer (any type) among spontaneously and ART-conceived children born in Denmark (1994–2014), Finland (1990–2014), Norway (1984–2015), and Sweden (1985–2015) and diagnosed with any cancer before age 18 years.** ART, assisted reproduction technology; CI, confidence interval.

No statistically significant differences for any cancer between ART-conceived and spontaneously conceived singletons (aHR 1.05, 95% CI 0.92 to 1.20, *p* = 0.48) or multiples (aHR 1.16, 95% CI 0.92 to 1.47, *p* = 0.22) were found.

## Risk of cancer after frozen-thawed embryo transfer

There were 48 cases of cancer in children born after FET (IR 30.1/100,000 person-years (Table 4). Children born after FET had a higher risk of any cancer compared both to children born after fresh embryo transfer (227 cases, IR 18.8/100,000 person-years, aHR 1.59, 95% CI 1.15 to 2.20, *p* = 0.005) and children born after spontaneous conception (aHR 1.65, 95% CI

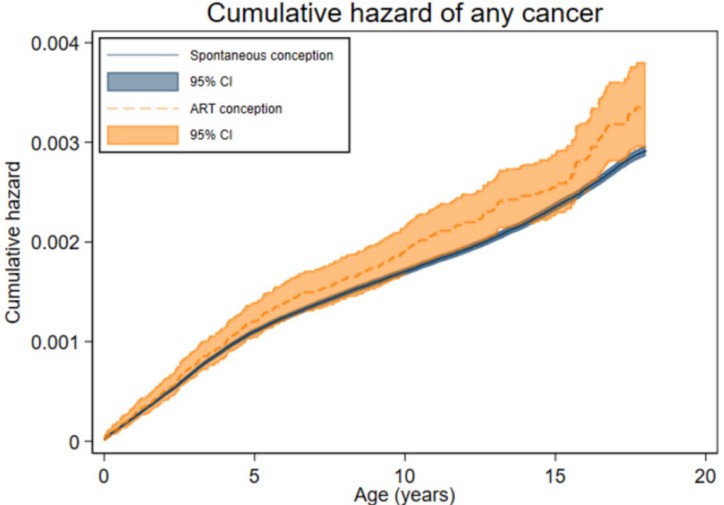

**Fig 3. Cumulative hazard of first cancer (any type) up to 18 years for spontaneously and ART-conceived children born in Denmark (1994–2014), Finland (1990–2014), Norway (1984–2015), and Sweden (1985–2015). Crude hazard ratio 1.13; 95% CI 1.01 to 1.26, *p* = 0.03.** ART, assisted reproduction technology; CI, confidence interval.

**Table 3. IR and risk of any cancer and type of cancer according to ICCC-3 categories before 18 years of age by first diagnosis and mode of conception in children born in Denmark, Finland, Norway, or Sweden (Denmark 1994–2014, Finland 1990–2014, Norway 1984–2015, and Sweden 1985–2015).**

| Cancer type (ICCC-3 category)[a] | ART N = 171,774 children N = 1,705,772 person-years | | | Spontaneous conception N = 7,772,474 children N = 97,027,051 person-years | | | ART vs. spontaneous conception | |
|---|---|---|---|---|---|---|---|---|
| | No. of children with cancer | IR | | No. of children with cancer | IR | | Crude HR (95% CI) p-value | Adjusted HR[b] (95% CI) p-value |
| | | Per 1,000 children | Per 100,000 person-years | | Per 1,000 children | Per 100,000 person-years | | |
| Any cancer (I–XII) | 329 | 1.92 | 19.29 | 16,184 | 2.08 | 16.68 | 1.13 (1.01 to 1.26) 0.03 | 1.08 (0.96 to 1.21) 0.18 |
| Leukemia (I) | 111 | 0.65 | 6.51 | 4,921 | 0.63 | 5.07 | 1.18 (0.98 to 1.43) 0.08 | 1.09 (0.89 to 1.33) 0.40 |
| Lymphomas (II) | 30 | 0.17 | 1.76 | 1,699 | 0.22 | 1.75 | 1.12 (0.78 to 1.61) 0.53 | 1.02 (0.71 to 1.49) 0.90 |
| Central nervous system tumors (III) | 87 | 0.51 | 5.10 | 4,080 | 0.52 | 4.20 | 1.20 (0.97 to 1.48) 0.10 | 1.22 (0.97 to 1.52) 0.09 |
| Neuroblastoma and other peripheral nervous cell tumors (IV) | 14 | 0.08 | 0.82 | 931 | 0.12 | 0.96 | 0.72 (0.42 to 1.21) 0.21 | 0.72 (0.42 to 1.24) 0.24 |
| Retinoblastoma (V) | 3 | 0.02 | 0.18 | 404 | 0.05 | 0.42 | NA[c] | NA[c] |
| Renal tumors (VI) | 17 | 0.10 | 1.00 | 841 | 0.11 | 0.87 | 0.99 (0.61 to 1.60) 0.96 | 1.07 (0.65 to 1.76) 0.79 |
| Hepatic tumors (VII) | 7 | 0.04 | 0.41 | 225 | 0.03 | 0.23 | NA[c] | NA[c] |
| Bone tumors (VIII) | 4 | 0.02 | 0.23 | 650 | 0.08 | 0.67 | NA[c] | NA[c] |
| Soft tissue sarcomas (IX) | 25 | 0.15 | 1.47 | 868 | 0.11 | 0.89 | 1.62 (1.09 to 2.41) 0.02 | 1.49 (0.98 to 2.27) 0.06 |
| Germ cell and gonadal tumors (X) | 7 | 0.04 | 0.41 | 667 | 0.09 | 0.69 | NA[c] | NA[c] |
| Epithelial tumors and melanoma (XI) | 22 | 0.13 | 1.29 | 812 | 0.10 | 0.84 | 2.00 (1.31 to 3.05) 0.001 | 1.89 (1.20 to 2.97) 0.01 |
| Other and unspecified tumors (XII) | <3[d] | <0.02[d] | <0.12[d] | 86 | 0.01 | 0.09 | NA[c] | NA[c] |

[a]US Department of Health and Human Services. National Institutes of Health. National Cancer Institute. International Classification of Childhood Cancer. ICCC Recode Third Edition ICD-O-3/IARC2017 [35].

[b]Adjusted for sex, plurality, year of birth, country of birth, maternal age at birth, and parity.

[c]Numbers too small (<10 in ART-conceived group) to calculate a stable estimate.

[d]These data not reported as exact numbers to protect patient confidentiality.

ART, assisted reproduction technology; CI, confidence interval; HR, hazard ratio; ICCC-3, International Classification of Childhood Cancer; IR, incidence rate; NA, not applicable.

1.24 to 2.19, $p = 0.001$). Singletons showed lower estimates than multiples (FET versus fresh embryo transfer, singletons aHR 1.49, 95% CI 1.01 to 2.18, $p = 0.04$; multiples aHR 1.91, 95% CI 1.04 to 3.50, $p = 0.04$ and FET versus spontaneous conception, singletons aHR 1.49, 95% CI 1.07 to 2.08, $p = 0.01$; multiples aHR 2.34, 95% CI 1.33 to 4.12, $p = 0.01$). In the FET group, the rates of macrosomia and major birth defects were: 31.3% versus 19.4%, 6.3% versus 4.3%, in the cancer and no cancer groups. Further adjustments for macrosomia or major birth defects only changed the results marginally (Table 4) as did adjusting for birth weight as a continuous

**Table 4. IR and risk of any cancer before 18 years of age in children conceived by FET, fresh embryo transfer, or spontaneous conception and born in Denmark, Norway, or Sweden[a] (Denmark 1994–2014, Norway 1984–2015, and Sweden 1985–2015).**

| | FET Children — All N = 22,630[b]; Singletons N = 18,872; Multiples N = 3,758; Person-years All N = 159,566; Singletons N = 124,839; Multiples N = 34,728 | | | Fresh embryo transfer Children — All N = 115,474; Singletons N = 83,623; Multiples N = 31,851; Person-years All N = 1,207,598; Singletons N = 804,450; Multiples N = 401,148 | | | Spontaneous conception Children — All N = 6,306,023; Singletons N = 6,142,874; Multiples N = 163,149; Person-years All N = 79,920,380; Singletons N = 77,896,756; Multiples N = 2,023,624 | | | FET vs. fresh embryo transfer | | | | FET vs. spontaneous conception | | | |
|---|---|---|---|---|---|---|---|---|---|---|---|---|---|---|---|---|---|
| | No. of children with cancer | Per 1,000 children | IR Per 100,000 person-years | No. of children with cancer | Per 1,000 children | IR Per 100,000 person-years | No. of children with cancer | Per 1,000 children | IR Per 100,000 person-years | HR (95% CI) p-value | aHR (95% CI) p-value | aHR (95% CI) p-value | aHR (95% CI) p-value | HR (95% CI) p-value | aHR (95% CI) p-value | aHR (95% CI) p-value | aHR (95% CI) p-value |
| All | 48 | 2.12 | 30.08 | 227 | 1.97 | 18.80 | 13,044 | 2.07 | 16.32 | 1.51 (1.11 to 2.07) 0.01 | 1.59[c] (1.15 to 2.20) 0.005 | 1.54[d] (1.11 to 2.14) 0.009; 1.55[e] (1.12 to 2.15) 0.09 | 1.59[f] (1.15 to 2.20) 0.005; 1.63[g] (1.17 to 2.26) 0.003 | 1.69 (1.27 to 2.24) <0.001 | 1.65[c] (1.24 to 2.19) 0.001 | 1.65[d] (1.24 to 2.19) 0.001; 1.62[e] (1.22 to 2.17) 0.001 | 1.87[f] (1.73 to 2.02) <0.001 |
| Singletons | 35 | 1.85 | 28.04 | 146 | 1.75 | 18.15 | 12,734 | 2.07 | 16.35 | 1.45 (1.0003 to 3.24) 0.05 | 1.49[h] (1.01 to 2.18) 0.04 | 1.42[i] (0.96 to 2.09) 0.08; 1.47[l] (1.00 to 2.16) 0.049 | 1.49[k] (1.01 to 2.18) 0.04; 1.51[l] (1.03 to 2.23) 0.04 | 1.55 (1.11 to 2.16) 0.01 | 1.49[h] (1.07 to 2.08) 0.01 | 1.49[i] (1.07 to 2.08) 0.02; 1.49[j] (1.07 to 2.08) 0.02 | 1.49[k] (1.06 to 2.07) 0.02 |
| Multiples | 13 | 3.46 | 37.43 | 81 | 2.54 | 20.19 | 310 | 1.90 | 15.32 | 1.80 (1.001 to 3.24) 0.05 | 1.91[h] 1.04 to 3.50 0.04 | 1.94[i] (1.06 to 3.54) 0.03; 1.93[j] (1.06 to 3.54) 0.03 | 1.91[k] (1.04 to 3.50) 0.04; 2.00[l] (1.09 to 3.65) 0.03 | 2.32 (1.33 to 4.05) 0.01 | 2.34[h] (1.33 to 4.12) 0.01 | 2.37[i] (1.34 to 4.18) 0.01; 2.36[j] (1.34 to 4.16) 0.003 | 2.34[k] (1.33 to 4.12) 0.01 |

[a] Without data from Finland because no information about frozen or fresh embryo transfer available in Finland.

[b] There were missing data on frozen or fresh embryo transfer in the ART group in 3,988/142,092 (2.8%).

[c] Adjusted for sex, plurality, year of birth, country of birth, maternal age at birth, and parity.

[d] Adjusted for sex, plurality, year of birth, country of birth, maternal age at birth, parity, and macrosomia ($\geq$4,000 g).

[e] Adjusted for sex, plurality, year of birth, country of birth, maternal age at birth, parity, and birth weight (continuous variable).

[f] Adjusted for sex, plurality, year of birth, country of birth, maternal age at birth, parity, and major birth defects [36].

[g] Adjusted for sex, plurality, year of birth, country of birth, maternal age at birth, parity, and embryo stage.

[h] Adjusted for sex, year of birth, country of birth, maternal age at birth, and parity.

[i] Adjusted for sex, year of birth, country of birth, maternal age at birth, parity, and macrosomia ($\geq$4,000 g).

[j] Adjusted for sex, year of birth, country of birth, maternal age at birth, parity, and birth weight (continuous variable).

[k] Adjusted for sex, year of birth, country of birth, maternal age at birth, parity, and major birth defects [36].

[l] Adjusted for sex, year of birth, country of birth, maternal age at birth, parity, and embryo stage.

aHR, adjusted hazard ratio; ART, assisted reproduction technology; CI, confidence interval; FET, frozen–thawed embryo transfer; HR, hazard ratio; IR, incidence rate.

variable instead of macrosomia. In the FET versus fresh embryo group, adjustment for embryo stage slightly strengthened the association (Table 4).

Risks of specific types of cancer in children born after FET versus fresh embryo transfer and versus spontaneous conception are presented in S5 Table. A higher risk was observed for leukemia in children born after FET (23 cases, IR 14.4/100,000 person-years) versus fresh embryo transfer (75 cases, IR 6.2/100,000 person-years) (aHR 2.25, 95% CI 1.38 to 3.68, $p = 0.001$) and in children born after FET versus spontaneous conception (aHR 2.22, 95% CI 1.47 to 3.35, $p < 0.001$). Further adjustment for macrosomia or major birth defects only attenuated the association slightly (S5 Table). The rates of any chromosomal aberration among children with leukemia were 0% in the FET, 4.0% in the fresh, and 2.2% in the spontaneous conception group. In the FET versus fresh embryo group adjustment for embryo stage slightly strengthened the association (S5 Table).

The HRs for covariates included in the regression analyses are illustrated in Figs 4 and 5 and S2.

## Discussion

The main finding in this large cohort study, based on nationwide registries and including 171,774 children born after use of any ART, was that while no increase in any childhood cancer was found after any ART, a higher risk was observed in children born after FET. The estimates were robust and changed only marginally after adjustment for relevant confounders. For specific cancer types, a significantly higher risk was found for epithelial tumors and melanomas in children born after ART versus spontaneous conception and for leukemia in children born after FET versus fresh embryo transfer and spontaneous conception. Further adjustments for either macrosomia, continuous birth weight, or birth defects only marginally attenuated these associations while adjusting for embryo morphology slightly strengthened the association. Associations for FET were weaker for singletons than for multiples.

The reason for a possible higher risk of cancer in children born after FET is not known. Each childhood cancer type has its own risk factor profile, but many childhood cancers are thought to derive from embryonic accidents and originate in utero [18]. High birth weight has been associated with higher childhood cancer risk, and epigenetic alterations have been

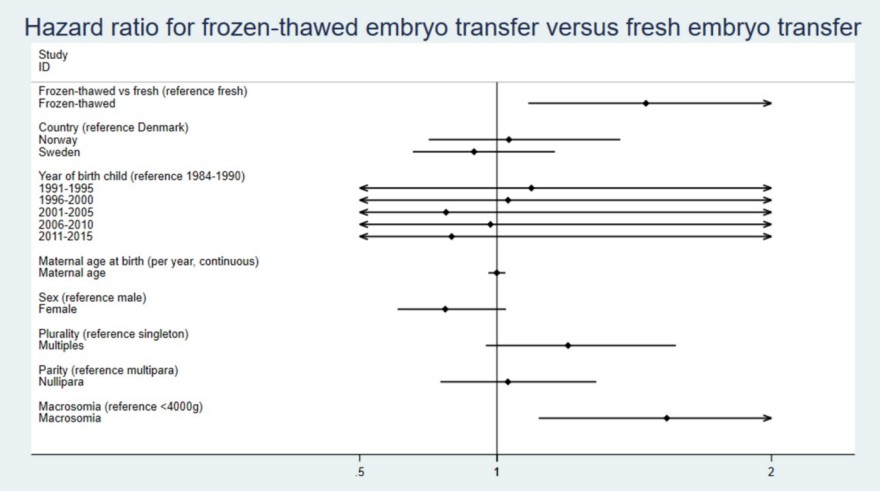

**Fig 4. HRs with 95% CI for independent covariates including macrosomia for risk of cancer in children born after FET versus fresh embryo transfer.** CI, confidence interval; FET, frozen-thawed embryo transfer; HR, hazard ratio.

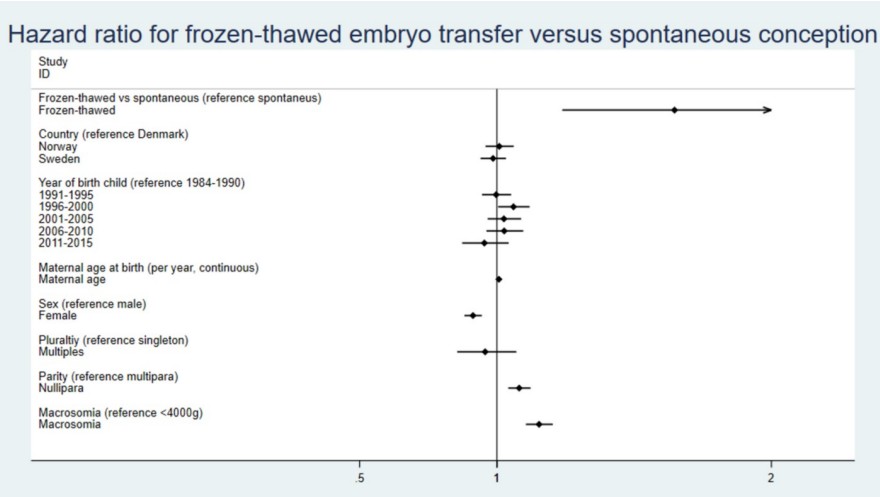

**Fig 5. HRs with 95% CI for independent covariates including macrosomia for risk of cancer in children born after FET versus spontaneous conception.** CI, confidence interval; FET, frozen-thawed embryo transfer; HR, hazard ratio.

proposed as a possible explanation [45–47]. Recent studies suggest changes in the epigenetic control in newborns after use of different ARTs [49,50]. A population-based US study found that among children with birth defects, particularly birth defects of chromosomal origin, those conceived via ART were at greater risk of developing cancer compared with spontaneously conceived children [23]. Although in our study, a major birth defect was an independent predictor of cancer in the analysis of children born after FET versus spontaneous conception, the association changed only marginally after adjustment for major birth defects, as did analyses with adjustment for macrosomia or birth weight. However, these analyses should be interpreted with caution due to the possibility of confounding from factors influencing both birth weight, major birth defects, and cancer risk [51,52]. Thus, further adjustment, separating chromosomal and non-chromosomal aberrations was not performed.

Higher risks of preterm birth, low birth weight, and birth defects in singletons after ART have been repeatedly found both in large cohort and registry-based studies and in systematic reviews and meta-analyses [53,54]. For children born after FET compared to children born after fresh embryo transfer, a lower risk of preterm birth and low birth weight, but a higher risk of macrosomia is apparent [48]. Studies on long-term outcomes in ART-conceived children are more limited. Divergent results have been published concerning childhood cancer after ART. Most large observational studies show similar cancer risk for children born after ART compared to the general population [20–23]. In a large cohort study in the United Kingdom, including 106,013 ART children [20], 108 children with cancer were identified, compared to 109.7 expected cancers (standardized IR 0.98, 95% CI 0.81 to 1.19). Higher risks were detected for certain malignancies such as hepatoblastoma and rhabdomyosarcoma. For children born after FET, the risk was similar to that in children born after fresh embryo transfer. Also, 2 earlier Nordic studies including 91,796 and 25,782 ART children, respectively, partly overlapping the present study, did not indicate any higher cancer risk in ART children (aHR 1.08, 95% CI 0.91 to 1.27 and aHR 1.21, 95% CI 0.90 to 1.63, respectively) [21,38]. In a large observational study in the USA, a slightly higher risk of cancer among children born after ART was observed (HR 1.17, 95% CI 1.00 to 1.36) [24]. The study identified 321 children with cancer in an ART population of 275,686 children, but no difference in risk was found for children born after FET. In contrast, a Danish population-based registry study found higher risk of any

childhood cancer after FET than after spontaneous conception, but the result was based on only 14 cases (HR 2.43, 95% CI 1.44 to 4.11) [22]. Studies on childhood cancer after ART were recently summarized in a systematic review [55], concluding that FET may be related to a higher risk of pediatric cancer. In an even more recent study from Israel, with a limited number of children, a higher risk of cancer was found in children born after fresh transfer [27]. The conflicting results may partly be due to limited study sizes with few events, differences in cancer registration, and various completeness of registries.

The main strengths of the present study are the large sample size, including unselected ART and spontaneously conceived populations born during a period of up to 3 decades in 4 Nordic countries and the use of high-quality validated population-based registries [56]. Individual data linkage between population-based registries made adjustments for potential confounders possible.

The main limitation is the number of children with cancer in the FET group. Although including a large cohort, this study cannot give a definite answer if FET is associated with an increased risk of cancer in childhood. It was not possible to include Finland in the FET analysis due to missing information on ART method. Further, there was also lack on information on emigration from Finland. Adjustment for race/ethnicity was not possible since registration on race/ethnicity is not allowed in the Nordic countries. It has been reported that non-white children and young adults might have lower rates of some childhood cancers [57]. The percentage of mother's country of birth being outside the Nordic countries was however low and similar in ART and spontaneous conception in an earlier publication from CoNARTaS [58].

Furthermore, all data are observational, and residual confounding by factors such as genetics, parental preconception health, and lifestyle cannot be excluded.

We were not able to exclude other medically assisted reproduction methods such as intrauterine insemination or ovulation induction from spontaneously conceived children. Although today such cycles, at least in Denmark are substantial, they only accounted for a small proportion of the spontaneous conception cohort. This misclassification might have attenuated the associations.

In the present study, only patients performing ART and delivering in their home countries are included. Although fertility tourism, meaning that patients go abroad or coming from abroad for fertility treatment today is rather common in some Nordic countries, this was uncommon during the study period. Such cycles are further impossible to correctly identify.

It might be argued that selecting the best quality embryos for fresh embryo transfer while cryopreserving less good quality could represent 2 morphologically different populations of embryos with different risks of any adverse outcome. Although numerous studies have found an association between embryo quality and pregnancy and live birth rates, there are at present no indication of more adverse outcome in children born from poor quality embryos [59]. In addition, more FET pregnancies were conceived after blastocyst transfers which were considered having higher quality than cleavage stage embryos and when adjusting for embryo stage as an indicator of embryo quality, the association between FET and cancer slightly strengthened. Furthermore, a vast majority of FET cycles were performed after a failed fresh cycle from the same oocyte retrieval and cryopreservation of surplus embryos while the freeze-all concept was hardly used. In fact, the FET population probably represents more good prognosis patients since it was possible to freeze embryos in the same cycle in addition to the fresh embryo transfer. Even though the present study included both children born after slow freezing of cleavage stage embryos and the more recent technique with vitrification of blastocysts, which could be associated with different risks, a recent large study comparing these techniques has not shown any major differences in perinatal outcome between these groups [60].

It is not clear if the results of this study can be broadly generalizable; however, the study population represents an unselected ART as well as spontaneously conceived cohort from 4 Nordic countries covering a long time period.

## Conclusions and further implications

In conclusion, while risk of any cancer was not higher in children born after use of ART, we found that children born after FET had a higher risk of childhood cancer than children born after fresh embryo transfer and spontaneous conception. The results should be interpreted cautiously based on the limited number of children with cancer. Although the absolute risk is low, these findings are important considering the increasing use of the freeze-all strategy. Future research should elucidate these results and the mechanisms behind.

## Supporting information

**S1 STROBE Checklist. STROBE, Strengthening the Reporting of Observational Studies in Epidemiology.**
(DOCX)

**S1 Text. Prospective analysis plan: Cancer in children born after frozen-thawed embryo transfer: A cohort study.**
(DOCX)

**S1 Fig. Flow chart of study population.**
(DOCX)

**S2 Fig. Hazard ratios with 95% confidence interval for independent covariates including major birth defects for risk of cancer in children born after frozen-thawed embryo transfer versus fresh embryo transfer and versus spontaneous conception.**
(DOCX)

**S1 Table. Data sources and standards.**
(DOCX)

**S2 Table. Main classification of cancer diagnosis groups according to the International Classification of Childhood Cancer (ICCC-3).**
(DOCX)

**S3 Table. Characteristics of study population by mode of conception defined as frozen-thawed embryo transfer, fresh embryo transfer, or spontaneous conception in children born in Denmark (1994–2014), Norway (1984–2015), and Sweden (1985–2015).**
(DOCX)

**S4 Table. Incidence rate of any cancer and type of cancer according to International Classification of Childhood Cancer (ICCC-3) categories before 18 years of age by first diagnosis and country of birth in children born in Denmark (1994–2014), Finland (1990–2014), Norway (1984–2015), and Sweden (1985–2015).**
(DOCX)

**S5 Table. Incidence rate and risk of specific type of cancer according to International Classification of Childhood Cancer (ICCC-3) by first diagnosis before 18 years of age in children born after frozen-thawed embryo transfer, fresh embryo transfer, or spontaneous conception in Denmark (1994–2014), Norway (1984–2015), and Sweden (1985–2015).**
(DOCX)

## Author Contributions

**Conceptualization:** Signe Opdahl, Mika Gissler, Anja Pinborg, Anna-Karina Aaris Henningsen, Aila Tiitinen, Liv Bente Romundstad, Christina Bergh, Ulla-Britt Wennerholm.

**Data curation:** Birgitta Lannering, Max Petzold, Signe Opdahl, Mika Gissler, Anne Lærke Spangmose, Christina Bergh, Ulla-Britt Wennerholm.

**Formal analysis:** Max Petzold, Christina Bergh, Ulla-Britt Wennerholm.

**Funding acquisition:** Birgitta Lannering, Signe Opdahl, Mika Gissler, Anja Pinborg, Aila Tiitinen, Christina Bergh, Ulla-Britt Wennerholm.

**Investigation:** Nona Sargisian, Birgitta Lannering, Signe Opdahl, Mika Gissler, Anja Pinborg, Christina Bergh, Ulla-Britt Wennerholm.

**Methodology:** Birgitta Lannering, Max Petzold, Signe Opdahl, Mika Gissler, Anja Pinborg, Christina Bergh, Ulla-Britt Wennerholm.

**Project administration:** Mika Gissler, Anne Lærke Spangmose, Christina Bergh, Ulla-Britt Wennerholm.

**Supervision:** Christina Bergh, Ulla-Britt Wennerholm.

**Validation:** Ulla-Britt Wennerholm.

**Writing – original draft:** Nona Sargisian, Birgitta Lannering, Christina Bergh, Ulla-Britt Wennerholm.

**Writing – review & editing:** Nona Sargisian, Birgitta Lannering, Max Petzold, Signe Opdahl, Mika Gissler, Anja Pinborg, Anna-Karina Aaris Henningsen, Aila Tiitinen, Liv Bente Romundstad, Anne Lærke Spangmose, Christina Bergh, Ulla-Britt Wennerholm.

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
