## [Editor Report · Decision Letter 0]

15 Mar 2022

Dear Dr Wennerholm, 

Thank you for submitting your manuscript entitled "Cancer in Children Born after Frozen-Thawed Embryo Transfer: A Cohort Study" for consideration by PLOS Medicine.

Your manuscript has now been evaluated by the PLOS Medicine editorial staff and I am writing to let you know that we would like to send your submission out for external peer review.

Please re-submit your manuscript within two working days, i.e. by Mar 17 2022 11:59PM.

Kind regards,

Beryne Odeny

PLOS Medicine

---

## [Decision Letter · Decision Letter 1]

28 Apr 2022

Dear Dr. Wennerholm,

Thank you very much for submitting your manuscript "Cancer in Children Born after Frozen-Thawed Embryo Transfer: A Cohort Study" (PMEDICINE-D-22-00822R1) for consideration at PLOS Medicine. 

[LINK]

In light of these reviews, I am afraid that we will not be able to accept the manuscript for publication in the journal in its current form, but we would like to consider a revised version that addresses the reviewers' and editors' comments. Obviously we cannot make any decision about publication until we have seen the revised manuscript and your response, and we plan to seek re-review by one or more of the reviewers. 

We expect to receive your revised manuscript by May 19 2022 11:59PM. Please email us (plosmedicine@plos.org) if you have any questions or concerns.

We look forward to receiving your revised manuscript. 

Sincerely,

Beryne Odeny, 

PLOS Medicine

plosmedicine.org

1) The Data Availability Statement (DAS) requires revision. For each data source used in your study: 

a) If the data are owned by a third party but freely available upon request, please note this and state the owner of the data set and contact information for data requests (web or email address). Note that a study author cannot be the contact person for the data.

b) If the data are not freely available, please describe briefly the ethical, legal, or contractual restriction that prevents you from sharing it. Please also include an appropriate contact (web or email address) for inquiries (again, this cannot be a study author).

2) Abstract:

a) Please ensure that all numbers presented in the abstract are present and identical to numbers presented in the main manuscript text.

b) Please quantify the main results (please present both 95% CIs and p values).

c) In the last sentence of the Abstract Methods and Findings section, please describe the main limitation(s) of the study's methodology.

3) Author summary – please include the main limitation(s) of the study 

4) Did your study have a prospective protocol or analysis plan? Please state this (either way) early in the Methods section. 

a) If a prospective analysis plan (from your funding proposal, IRB or other ethics committee submission, study protocol, or other planning document written before analyzing the data) was used in designing the study, please include the relevant prospectively written document with your revised manuscript as a Supporting Information file to be published alongside your study and cite it in the Methods section. A legend for this file should be included at the end of your manuscript. 

5) Please add the following statement, or similar, to the Methods: "This study is reported as per the Strengthening the Reporting of Observational Studies in Epidemiology (STROBE) guideline (S1 Checklist)."

a) Thank you for providing the STROBE checklist. When completing the checklist, please use section and paragraph numbers, rather than page numbers.

6) Statistical analysis

a) Please include adjustment variables. 

b) Did you adjust for clustering at the parent level i.e., where 2 or more children are born via ART or spontaneous conception in a family 

7) Please provide p values in addition to 95% CIs in the main text and tables

8) Please present and organize the Discussion as follows: a short, clear summary of the article's findings; what the study adds to existing research and where and why the results may differ from previous research; strengths and limitations of the study; implications and next steps for research, clinical practice, and/or public policy; one-paragraph conclusion.

9) In light of the study limitations, please avoid overstating your conclusions

10) Please remove the ‘Data sharing sentence” and “Transparency declaration” from the end of the main text. In the event of publication, this information will be published as metadata based on your responses to the submission form.

11) References:

a) Please select the PLOS Medicine reference style in your citation manager. In-text reference call outs should be presented as follows noting the absence of spaces within the square brackets, e.g., "... services [1,2]."

b) References should have six names before et al. For those with more than six names, please ensure that et al., is inserted after six names

c) Please ensure that journal name abbreviations consistently match those found in the National Center for Biotechnology Information (NCBI) databases. https://journals.plos.org/plosmedicine/s/submission-guidelines#loc-references. 

d) Ref #1, 14, 30, 31, 32, 35, 36 are incomplete – please include the article titles.

e) Please update ref #34 or delete if not published.

Comments from the reviewers:

Reviewer #1: I confine my remarks to statistical aspects of this paper. The general approach is fine, but I do have some issues to resolve before I can recommend publication.

I especially commend the authors for building models based on science rather than on anything like forward, backward, or stepwise selection and for including sensitivity analyses.

One general issue is that the authors have an entire population, not a sample. Many statisticians (including me) feel that this makes inference (and, therefore, p values and CIs) irrelevant. There is no population to infer to. Others posit some "super population". I don't find this persuasive (for one thing, how is your population a random sample from that?) but, if the authors want to do that, I won't forbid publication. But this needs to be dealt with. One advantage of my approach is that it downplays p values altogether, which is a good thing because they are used far too much.

p. 9 line 210-211 - Complete case analysis is only OK if the amount of missing data is either very small or missing completely at random. I realize this would go in the Results section, but it could be mentioned here. (This is really a style issue, the editors may have a preference). However, even in results, I only saw one mention of missing data, in the footnote to a table. So ... Please add something about this in the text.

Table 1 - there's nothing really wrong here, but it's a bit hard to read because of the number of columns and their organization. Are the number of children and number with cancer both needed? I think you can easily drop number with cancer, since you have the total and the %. Do you need both number of kids and PY? And ... Maybe organize the columns by subject and then group rather than the reverse? That is, have the three incident rates per 1000 children next to each other, and the incidence rate per PY next to each other? Again, this is just to make it easier for the reader to make the most relevant comparisons.

 or maybe table 1 could be organized like Table 3 with the sample sizes in the header?

Figure 1 - It would be better to combine the two panels and use a line for each group, rather than a series of bars. Also, the labels on the y-axis could be made horizontal (but that is not a big deal).

Figure 2 - I would remove the lines for the CIs (especially considering my first point) I might also remove the shading. And same thing with the y-axis labels (This is easy to change in R, but I don't know about Stata).

Overall, though, a very good job. 

Reviewer #2: I wrote a review of this analysis in December, 2021, when it was submitted to the Annals of Internal Medicine (see attachment). Although some small changes have been made, this version is essentially the same as that manuscript, therefore much of my original review and comments are still valid. The primary comment regarding this version is that the conclusion (in the abstract, and the Discussion) is that FET is associated with a higher risk of childhood cancer, yet the small sample size is the first comment in the limitations (page 12, line 305). Although the authors criticize the Hargreave paper (JAMA 2019; 322(22):2203-2210) for only having 14 cancer cases among their FET cohort, many of the analyses in this manuscript are based on very small sample sizes--Table 2 includes 5 cancer types based on <10 cancer cases, and 7 cancer types based on <20 cancer cases. My primary recommendation (in addition to those from my original review) would be to limit the analyses to only all cancers and to leukemia. 

This analysis should also be limited to residents of each of the study countries who received their ART treatment within their home country; the lack of emigration data for Finland is also problematic.

Reviewer #3: This is a very important study, dealing with an exposure that is relatively new, still it is becoming more prevalent. The study is based on a very large sample size, which is a major advantage, as mentioned by the authors, with a long follow-up time, which is suitable to study such outcomes.

I do have a few suggestions and comments: 

In the Abstract- Please add the rate of ART in this cohort (not only the n`s).

In the Abstract- please explain why there are 2 follow-up periods mentioned.

Please add in the Abstract how many cases were diagnosed among the fresh embryos, and be consistent with the reporting style.

In the Abstract- Obviously the models did not adjust for both macrosomia and birthweight, please clarify.

In the abstract- "all children born 1994-2014…" should be- all children born between the years ….

The sentence in line#120 ("In many countries, the number of FET-conceived children has now exceeded the number born after fresh embryo transfer.") should be transferred to line#104.

The following sentence is unclear: "Cancer in childhood is heterogeneous especially concerning morphology" , line#124. What does it mean cancer is heterogeneous? In what way?

In the Methods section- How were the co-variables in the models selected? 

In the results section- lines#250-262, several multivariable models are mentioned- what were included in these models? 

What the authors performed is not a sensitivity analysis in sub-populations, rather they tested several models.

What does it add to show the results by country? 

Indeed, as he authors mentioned- there is a possibility of selection bias, as the best quality embryos may have been selected for fresh embryo transfer. The authors should consider using propensity scoring analysis, to possibly account for this selection bias.

Did the authors account for siblings in the cohort, and how? 

Reviewer #4: This is a very well-written manuscript using a robust population-based approach to evaluate associations between frozen-thawed embryo transfer (FET) and cancer in children. The analyses were comprehensive and clear, and the findings are notable. I do have a few questions for the authors to consider.

Methods: While I have no objections to the choice of covariates for the analysis, I was curious about the rationale for the selection of certain variables. Was this done based on previous assessments, the construction of a DAG, or some other strategy? I think providing some rationale for variable selection in the Methods section would be helpful.

Methods: I appreciate that the authors considered the role of birthweight and birth defects on their results. I have two questions related to this. First, how was information on birth defects obtained? Were similar linkages done using congenital anomaly registries, or was this information obtained from birth records. If the latter, this is a potential limitation for this analysis as birth records are neither sensitive nor specific for the assessment of birth defects (see PMID: 27859434). I think this point must be addressed. Second, if the authors propose that these conditions could be important in the potential association between FET and cancer in children, it seems as if they would be on the causal pathway rather than factors that need to be considered as confounders. I think even describing the prevalence of macrosomia and birth defects in those exposed to FET with and without cancer would be interesting. Ultimately, can the authors justify this approach (i.e., thinking of phenotypes on the causal pathway as confounders)?

Methods: Use of the ICCC-3 for cancer classification is a strength. Can the authors confirm if these classifications were done for older years by the respective cancer registries (I don't think the ICCC-3 was used until the 2000s) - or did the authors use an algorithm to code these cancers? Either way, can the authors include this information in the Methods?

Results: The finding related to FET and leukemia is interesting. To that point, can the authors explain this result a bit more? For example, were these cases all ALL or AML? As trisomy 21 is a strong risk factor for pediatric leukemia, were any of these children with leukemia also diagnosed with trisomy 21? I know the authors adjusted for birth defects, but I think this particular congenital anomaly should be considered.

[LINK]

---

## [Decision Letter · Decision Letter 2]

27 Jun 2022

Dear Dr. Wennerholm,

Thank you very much for submitting your manuscript "Cancer in Children Born after Frozen-Thawed Embryo Transfer: A Cohort Study" (PMEDICINE-D-22-00822R2) for consideration at PLOS Medicine. 

[LINK]

In light of these reviews, I am afraid that we will not be able to accept the manuscript for publication in the journal in its current form, but we would like to consider a revised version that addresses the reviewers' and editors' comments. Obviously we cannot make any decision about publication until we have seen the revised manuscript and your response, and we plan to seek re-review by one or more of the reviewers. 

We expect to receive your revised manuscript by Jul 18 2022 11:59PM. Please email us (plosmedicine@plos.org) if you have any questions or concerns.

We look forward to receiving your revised manuscript. 

Sincerely,

Beryne Odeny, 

PLOS Medicine

plosmedicine.org

1. Please pay particular attention to reviewer #3’s remaining concerns regarding your statistical approach. We will not be able to proceed until you respond to this.

2. Please report p <0.001 instead of p <0.01

3. Abstract

a. Change subheading to “Methods and findings” not “Method and findings”

b. Last sentence of the “Methods and findings” should read, “The main limitation of this study is the small number of children with…” Please update the author summary as well.

4. Please remove subheadings from the Discussion section

Comments from the reviewers:

Reviewer #2: The authors have been responsive to most of the issues raised in the review. Two issues remain: the use of Marsal et al, 1996 birthweight reference to characterize the study children as SGA or LGA; and Cox proportional hazards models based on <5 FET children. For the birthweight reference, since neither SGA nor LGA is used in the models (with macrosomia and LBW used instead), the authors should eliminate the SGA and LGA data and related text, and in future analyses, use a better reference when generating these cut-offs. For the models using <5 FET children in the 7 out of 9 cancer types (Table S5), I still feel strongly than even giving the crude HRs is somewhat misleading, although the authors have excluded the adjusted HRs. I understand their desire to include absolute numbers for the possible inclusion in future meta-analyses. In closing, I want to commend the authors for this ambitious study, and for its contribution to the field.

Reviewer #3: While most of my comments have been addressed in the revised submission, I still have concerns about the selection of variables in the multivariable analysis.

Selecting variables based on the literature is not the correct and may not be relevant in this study population. The selection should be based on testing the confounding effect,

model fit, and testing possible collinearity between co-variables. 

This is not mentioned and the methods not detailed.

Additionally, lack of data regarding sperm morphology does not make propensity score analysis irrelevant or impossible, and still I recommend adding this analysis to strengthen your findings.

[LINK]

---

## [Decision Letter · Decision Letter 3]

21 Jul 2022

Dear Dr Wennerholm, 

On behalf of my colleagues and the Academic Editor, Dr. Jenny E Myers, I am pleased to inform you that we have agreed to publish your manuscript "Cancer in Children Born after Frozen-Thawed Embryo Transfer: A Cohort Study" (PMEDICINE-D-22-00822R3) in PLOS Medicine.

PRESS

Sincerely, 

Beryne Odeny 

PLOS Medicine